# Green Space Visits and Barriers to Visiting during the COVID-19 Pandemic: A Three-Wave Nationally Representative Cross-Sectional Study of UK Adults

Hannah Burnett *, Jonathan R. Olsen and Richard Mitchell

MRC/CSO Social and Public Health Sciences Unit, University of Glasgow, Glasgow G12 8QQ, UK; jonathan.olsen@glasgow.ac.uk (J.R.O.); richard.mitchell@glasgow.ac.uk (R.M.)
* Correspondence: h.burnett.1@research.gla.ac.uk

**Abstract:** Green spaces have been found to promote physical activity, social contact, and mental wellbeing, however, there are inequalities in the use and experience of green spaces. The United Kingdom's (UK) response to the COVID-19 pandemic imposed very substantial changes on its citizens' lives which could plausibly affect their willingness to visit green spaces. These sudden lifestyle changes severely affected the population's mental health, leading to a greater dependency on the positive influence of nature in reducing stress and improving mood. Whilst early cross-sectional evidence suggested an increased orientation to nature and visits to green spaces as a response to COVID-19 'lockdowns', there is little longitudinal evidence about how sustained and equal these changes may have been. This study explored green space visits, barriers to visiting, and the inequalities of both of those over an entire year of the pandemic in the UK. Three waves of nationally representative cross-sectional surveys were administered by YouGov in April 2020, November 2020, and April 2021 (N = 6713). Data included reported visits to green spaces and, for those with no or infrequent visiting, perceived barriers including those plausibly related to the risk of COVID-19. Green space visits increased over the year as lockdown restrictions were relaxed; 68% of respondents reported green space visits in April 2021, compared with 49% in April 2020. However, the socio-economic inequalities in use were sustained and increased. COVID-19 related barriers fell over time, but there were indications of increased interest in green spaces among younger people. Further action is required to ensure that the positive impacts of green spaces are experienced equally, and that good quality green space is accessible to all.

**Keywords:** green space; COVID-19; inequalities; barriers; health



## 1. Introduction

In late 2019, a virulent novel coronavirus was identified. Commonly known as COVID-19, the disease is spread from an infected person's mouth or nose in small liquid particles [1]. To reduce the rate of infection, restrictions on people's interaction and movement have been implemented within and between many countries around the world. At their most extreme, these restrictions included 'lockdowns' requiring most citizens to stay at, or close to, home. The precise timing, duration, and severity of restrictions varied according to the stage of the epidemic in each country.

The first wave of COVID-19 infections took hold in the UK in early 2020. The first lockdown started on 23 March 2020. Most people were only permitted to leave their homes for 'essential' reasons including access to health care, food, and to undertake one form of outdoor exercise [2]. In the subsequent two years, restrictions on movement and social interaction waxed and waned depending on the number of COVID-19 cases and the emergence of new variants. The next substantial set of UK restrictions was imposed between January and April 2021 when another 'stay-at-home' order was implemented

and individuals were prohibited from social mixing. In contrast to March 2020 however, restrictions were not placed on the form and frequency of outdoor exercise.

The pandemic and its lockdown restrictions have had a substantial impact on individual behaviours, routines, and habits worldwide [3,4]. Research has highlighted profound effects on population health. In addition to the morbidity and mortality from COVID-19 itself, both reduced access to health care and the experience of the pandemic has resulted in considerable morbidity. Studies in the UK, for example, found that the mental health of adults followed an 'up and down' cycle coinciding with periods of national lockdown and high COVID-19 case rates [5,6]. This negative impact on mental health has been felt internationally as well [7–10]. For example, in a study undertaken in Germany, perceived stress increased across all age groups over the six weeks following the mid-March 2020 COVID-19 outbreak [7]. The exact mental health burden of the pandemic is still too early to measure, with many countries experiencing new waves of virus cases [8]. Furthermore, the impacts of both COVID-19 and its wider consequences have not been even across society. The pandemic appears to have exacerbated pre-existing socio-demographic and geographic health inequalities. In the UK, for example, case rates grew faster and had higher peaks within more deprived areas [11,12].

Visiting green spaces has been shown to benefit mental and physical health outcomes. In the UK, considerable media attention was lavished on how green spaces could be a helpful resource during the pandemic and might mitigate the negative impacts on mental health [13,14]. Cross-sectional research suggested that during the early phases of the COVID-19 pandemic, the frequency of visits to green spaces increased and that this may have had a positive impact on mental health [15–18]. However, as restrictions were lifted and then reimposed, and the population adjusted to the 'new normal', it is probable that visit levels continued to change. Moreover, increased visits to green spaces and their perceptions of mental health benefits were not experienced equally across socio-demographic groups [17].

There are considerable social, economic, and gender inequalities in visits to green spaces; less advantaged populations are far less likely to do so [19–21]. Barriers to the use of green spaces are socially and spatially unequal. Not everyone has good access to a suitable space, or the time and inclination to visit it. This paper is concerned with measuring green space use from an environmental justice and inequalities perspective by exploring the differences according to socio-economic characteristics at a neighbourhood and individual level [22,23]. This paper aimed to provide insights that may add to the existing environmental justice framework as applied to green spaces, with a focus on the use and barriers rather than access and quality [24]. Whereas much environmental justice research explores inequalities in exposure to pathogenic environments, our work focuses on justice in both access to, and use of, a salutogenic environment. We explore how the COVID-19 pandemic may have exacerbated environmental injustice, particularly for those of differing social grades/occupations. It is important to explore environmental injustice alongside the "equigenic hypothesis" [25] which suggests that some environments (such as natural environments) can support the health of the less advantaged as much as, or even more than, the more advantaged [26,27]. By exploring green space visits and barriers during the pandemic, we can strengthen our understanding of who is not using green spaces, why, and how the pandemic impacted existing inequalities.

COVID-19 added several other pandemic-specific barriers including the desire and/or urgent need to socially distance, a sense of severe vulnerability to the virus, and sharp divides between who is, and is not, able or required to work at home or has access to outdoor space there [28–30]. These barriers are of particular interest in this paper. As vaccination proceeded, the population 'got used' to COVID-19, and as the public health response phased up and down in reaction to case-numbers and variants, the significance of COVID-19-specific barriers could plausibly change. A recent study in Korea found that individuals with decreased visits to green spaces between September–December 2020 had 2.06 higher odds of probable major depression at the time of the survey compared to those

whose visits had increased or stayed the same [31]. This emphasises the importance of researching the barriers to green space use during the pandemic by sociodemographic characteristics and exploring how the barriers changed over its first year.

Further research is required to describe whether the increased visit numbers were sustained as the COVID-19 pandemic continued, how barriers to visiting changed following subsequent easing and reintroduction of COVID-19 measures, and the inequalities in these. There is a lack of literature in this area, with no studies to date focusing on the change in green space use and barriers in the UK over the first year of the pandemic. It is important to explore this gap, both in order to understand how the experience of the UK relates to that in other countries, and also to promote resilient green spaces and wider societies in the face of future pandemics [32]. The research objectives of this paper were, therefore:

1. To describe variation in green space visiting during the COVID-19 pandemic.
2. To describe reported barriers to green space visits during the COVID-19 pandemic.
3. To explore variation in green space visits and barriers over time and by sex, age, and socio-economic position.

## 2. Materials and Methods

### 2.1. Survey Sample

Three waves of a repeat cross-sectional survey were administered by YouGov between April 2020 and April 2021 [33]. Each wave was drawn from YouGov's UK/GB Omnibus of 800,000 panellists. Respondents were selected at random from this panel by YouGov and then sent a survey link to complete. All three waves were nationally representative of the UK population when weightings were applied. The survey waves are described below.

**Wave 1:** The first wave was administered between 30 April and 1 May 2020, with a sample of 2252 UK adults aged 18+. At that point, the UK population was in the first 'lockdown', also known as the 'stay at home phase'. From 23 March 2020, people were only permitted to leave home for limited purposes, including collecting medicines, doing essential shopping, and doing one form of exercise per day [2]. When wave 1 was administered, the same lockdown restrictions were implemented across constituent nations of the UK.

**Wave 2:** The second survey wave was administered on 26 November 2020, with a sample of 2246 UK adults. When this wave was undertaken, COVID-19 policies and restrictions differed among the constituent nations of the UK. England was in a winter lockdown, with the population asked to stay at home and only leave for limited reasons such as education, essential shopping, exercise, health care, or to care for vulnerable people [34]. Wales was just out of a strict lockdown, with gyms, schools, and restaurants being reopened [35]. Scotland was operating localized lockdowns with almost half of its population under strict restrictions, including a ban on indoor household socialising and only essential shops being open [36].

**Wave 3:** The third wave was administered from 29–30 April 2021, exactly one year after wave 1, with a sample of 2215 UK adults. At this time, lockdown restrictions had been eased across the UK. Non-essential shops had reopened, outdoor gatherings for up to six people were allowed, and the population was able to travel outside of their local area [37].

### 2.2. Survey Content

Wave 1 was initially designed and implemented as a one-off cross-sectional survey but, as the COVID-19 pandemic persisted and evolved, and the UK introduced further restrictions, the two subsequent surveys were commissioned. Some question wording, therefore, differed very slightly between wave 1, and waves 2 and 3 (Table 1).

At every wave, respondents were asked about their green space visiting frequency. Green spaces were defined as 'places where you can see and experience plants, trees, and nature outside of the household (e.g., public parks, sports fields, agricultural land, woodlands, coastal paths, and nature reserves)'. Those that had not visited green spaces or had visited infrequently (once every 2 weeks or once in the last 4 weeks in waves 2 and 3)

were asked about the barriers to visiting green spaces and reasons for their non or low-frequency visiting (Table 1). The surveys covered a range of reported barriers to the use of green spaces, with 12 barriers included in wave 1 and 15 barriers in waves 2 and 3. In this analysis, we focused largely on those particularly relevant to COVID-19 and the lockdown restrictions. These were reported as: 'worried about social distancing in green space', 'green spaces are too busy', 'fear for health when outdoors (i.e., contracting COVID-19)', 'member of household/individual at risk of being severely affected by COVID-19', 'using an outdoor space at home instead', and 'not interested'.

**Table 1.** The survey themes, question wording (by wave), and response categories.

| Themes | Wave 1 Questions | Waves 2 and 3 Questions | Response Categories |
|---|---|---|---|
| **Visitation frequency** | Have you visited a green space since the movement restrictions have been enforced in the UK? (i.e., since 23 March 2020). | Have you visited ANY green spaces in the last 4 weeks | 'Yes, I have' 'No, I have not' |
| **Barriers to visits** | [If the respondent had not visited green space since lockdown was implemented] Which, if any, of the following are reasons for you not visiting green spaces since the restrictions were introduced (i.e., 23 March 2020)? (Please select all that apply)<br><br>1. I am worried that I will not be able to socially distance from others in these spaces (i.e., remain 2 metres away)<br>2. Green spaces are much busier now<br>3. I fear for my health when I go outdoors (i.e., contracting Coronavirus (COVID-19)<br>4. I/a member of my household is at higher risk of being severely affected by Coronavirus (COVID-19)<br>5. I am using an outside space at my home (e.g., garden) instead<br>6. I am not interested in visiting green spaces | [If the respondent had not visited green space or had visited infrequently in the last 4 weeks] You previously said you have not regularly visited a green space in the last 4 weeks . . . Which, if any, of the following are your reasons for this? (Please select all that apply)<br><br>1. I am worried that I will not be able to socially distance from others in these spaces (i.e., remain 2 metres away)<br>2. Green spaces are too busy for me (e.g., I can't enjoy them when they are crowded, they aren't peaceful enough, I feel uncomfortable surrounded by that many people etc.)<br>3. I fear for my health when I go outdoors (i.e., contracting Coronavirus (COVID-19))<br>4. I/a member of my household is at higher risk of being severely affected by Coronavirus (COVID-19)<br>5. I am using an outside space at my home (e.g., garden etc.) instead<br>6. I am not interested in visiting green spaces | 'Yes' 'No' |

Individual demographic and socio-economic (henceforth socio-demographic) characteristics, known to be associated with green space visiting, were also collected. These were: sex (male, female); age group (18–24 years, 25–64 years, 65+ years); and social grade (higher social grade, lower social grade) categorised by YouGov using combined occupational social grade categories (Table S1). Higher social grades included non-manual workers, such as senior managers, whilst lower social grades included all manual workers, such as labourers [38]. The demographic and socio-economic variables were consistent across all survey waves.

*2.3. Statistical Analyses*

Multiple statistical analyses were conducted to cover each of the research objectives. Firstly, to explore general patterns of green space use and barriers, descriptive statistics were run in R (version 3.5.1) [39]. This included calculating the weighted count and proportion of respondents who had visited green spaces and reported each barrier. These were explored by sex, age, social grade, and survey wave. Cross-tabulations with Pearson's $X^2$ were also used to test for significant differences between the groups.

Next, multiple binary logistic regression analyses were conducted to assess the associations between visiting green space, survey wave, the socio-demographic variables, and the reporting of each barrier. Separate models were run for each barrier.

Finally, interaction terms were added in order to investigate change over time (i.e., between waves) in relationships between the socio-demographic variables, survey wave, and green space visits, and between the socio-demographic variables, survey wave, and each

reported barrier. The significance of each interaction was assessed via Wald tests and only those which reached a threshold of $p < 0.05$ were examined in detail. Predicted probabilities were derived to ease the interpretation of the significant interaction terms. Weightings were applied during all analyses to ensure the sample was representative of the UK adult population. Sample weights were calculated and provided by YouGov [40].

## 3. Results

### 3.1. Descriptive Statistics

In wave 1 (April 2020), 49% of respondents reported that they had visited green spaces in the 4 weeks prior to the survey. This increased to 65% of respondents in wave 2 (November 2020), and 68% in wave 3 (April 2021) (Table 2, Figure 1). The differences in socio-demographic characteristics can be found in Table 2 and Figure 2.

In all survey waves, the most common reason for not visiting green space was "I am using an outside space at my home (e.g., garden) instead" (wave 1: 47%, wave 2: 26%, and wave 3: 32% (Table 2)). In wave 1, the second most commonly reported barrier was, "I fear for my health when I go outdoors (i.e., contracting Coronavirus (COVID-19)" (27%). In wave 2, the second most commonly reported barrier was "I/a member of my household is at higher risk of being severely affected by Coronavirus (COVID-19)" (15%). By wave 3, the second most common reason was "Green spaces are too busy for me (e.g., I can't enjoy them when they are crowded, they aren't peaceful enough, I feel uncomfortable surrounded by that many people, etc.)" (18%).

**Table 2.** Proportions visiting green spaces or reporting barriers to doing so.

| | | Those Who Visited Green Space in the Previous 4 Weeks | Those Who Either Did Not Visit Green Space in the Previous 4 Weeks or Did So Infrequently ^ | | | | | |
|---|---|---|---|---|---|---|---|---|
| | | % (n) | % (n) | Worried about Social Distancing in Green Spaces | Green Spaces Are Too Busy | Fear for Health When Outdoors (i.e., Contracting COVID-19) | Member of House-hold/Individual at Risk of Being Severely Affected by COVID-19 | Using an Outdoor Space at Home Instead | Not Interested in Visiting Green Spaces |
| **Wave** | | | | | | | | | |
| **1 April 20** | % | 49% (1086) | 50% (1123) | 25% | 9% | 27% | 26% | 47% | 8% |
| **2 Nov 20** | % | 65% (1421) | 45% (1020) | 14% | 9% | 14% | 15% | 26% | 9% |
| **3 April 21** | % | 68% (1479) | 45% (987) | 14% | 18% | 10% | 8% | 32% | 10% |
| **Sex \*** | | | | | | | | | |
| **Male** | % | 61% (1934) | 45% (1476) | 17% | 10% | 16% | 16% | 30% | 12% |
| **Female** | % | 61% (2052) | 48% (1654) | 19% | 13% | 19% | 18% | 39% | 7% |
| **Age group \*** | | | | | | | | | |
| **18–24** | % | 60% (411) | 46% (340) | 19% | 14% | 15% | 12% | 22% | 14% |
| **25–64** | % | 62% (2685) | 46% (2031) | 18% | 12% | 16% | 14% | 30% | 8% |
| **65+** | % | 58% (890) | 49% (758) | 18% | 10% | 23% | 27% | 54% | 9% |
| **Social grade \*** | | | | | | | | | |
| **Higher social grade** | % | 66% (2492) | 41% (1578) | 20% | 12% | 18% | 16% | 36% | 9% |
| **Lower social grade** | % | 53% (1494) | 54% (1552) | 16% | 11% | 17% | 18% | 33% | 10% |

\* Responses by demographic variables combined for all three waves, Chi$^2$ *p*-values < 0.05. All N's were weighted to account for survey response bias. ^ Infrequently defined as once every 2 weeks or once in the previous 4 weeks.

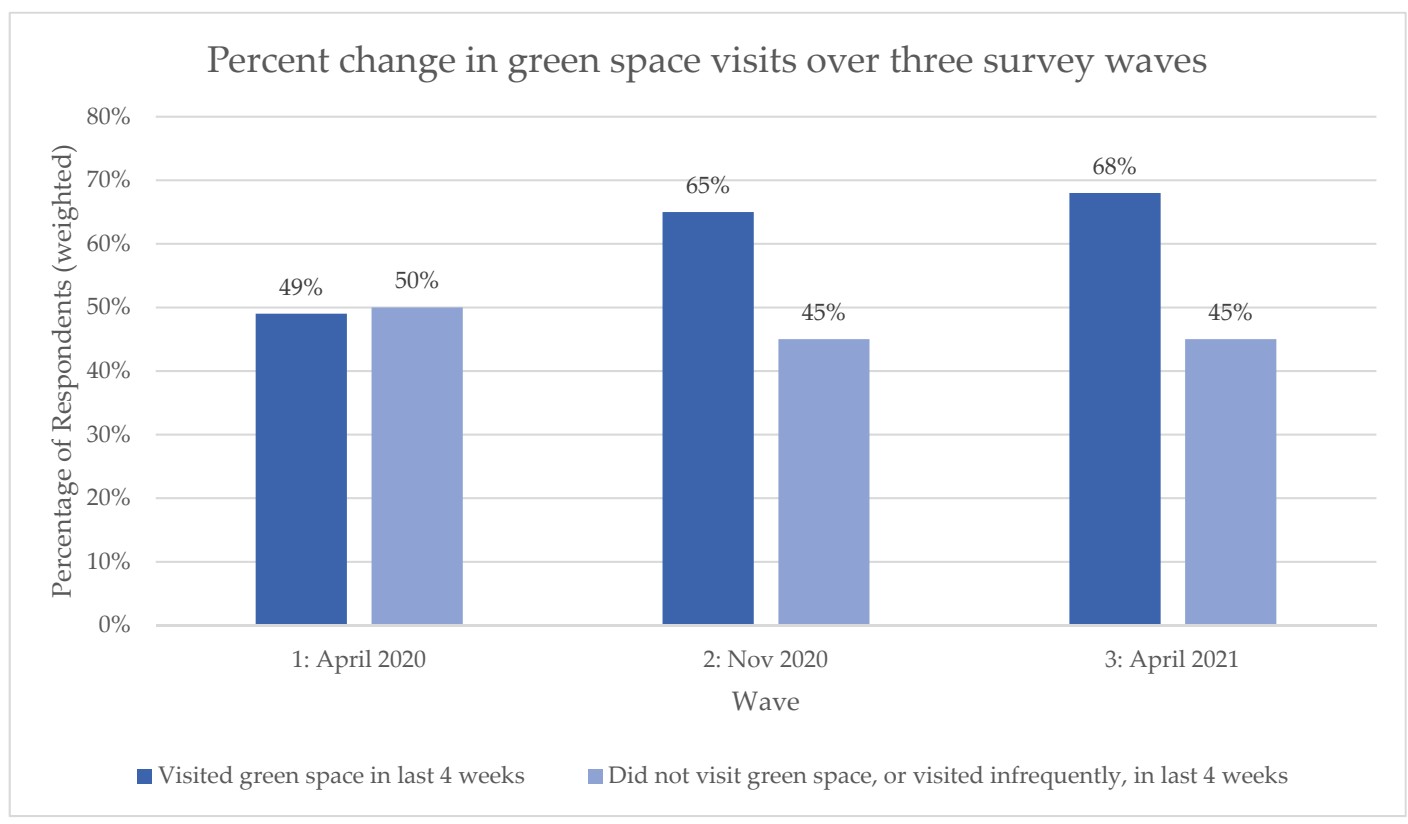

**Figure 1.** Change in the percentage of respondents stating visits to green space over the three survey waves (all significant Chi$^2$, *p*-values < 0.05).

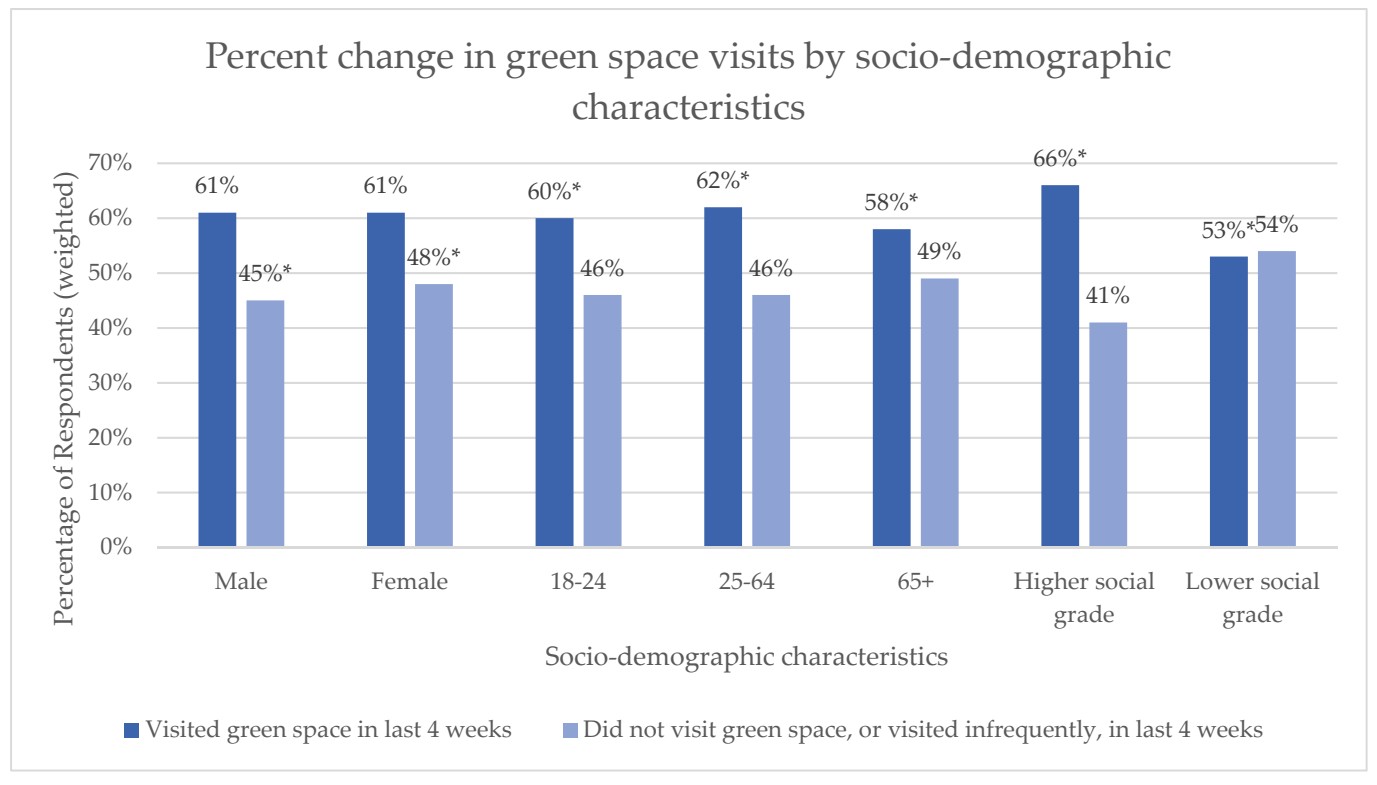

**Figure 2.** Change in the percentage of respondents stating visits to green space by socio-demographic characteristics (* Chi$^2$ *p*-values < 0.05).

### 3.2. Variation in Green Space Visits and Barriers to Visits by Wave

After adjustment for socio-demographic characteristics, the odds of respondents reporting visiting green spaces were significantly higher in wave 2 (OR: 1.28 (95% CI: 1.15–1.44)) than in wave 1 (Table 3). By wave 3, they were significantly higher than both waves 1 and 2 (OR: 2.25 (95% CI: 1.99–2.55)).

**Table 3.** Odds ratios (95% CI) from logistic regression models predicting either visiting green space or reported barriers to visiting.

| | Visited Green Space in Previous 4 Weeks | Barriers Reported by Those Who Either Did Not Visit Green Spaces in the Previous 4 Weeks or Did So Infrequently | | | | | |
|---|---|---|---|---|---|---|---|
| | | Worried about Social Distancing in Green Spaces | Green Spaces Are Too Busy | Fear for Health When Outdoors | Member of Household/Individual at Risk of Being Severely Affected by COVID-19 | Using an Outdoor Space at Home Instead | Not Interested in Visiting Green Spaces |
| *Wave* | | | | | | | |
| 1 (ref) (April 20) | 1.00 | 1.00 | 1.00 | 1.00 | 1.00 | 1.00 | 1.00 |
| 2 (November 20) | 1.28 (1.15–1.44) *** | 0.51 (0.41–0.64) *** | 1.06 (0.79–1.44) | 0.43 (0.34–0.53) *** | 0.49 (0.40–0.62) *** | 0.39 (0.33–0.47) *** | 1.20 (0.88–1.64) |
| 3 (April 21) | 2.25 (1.99–2.55) *** | 0.50 (0.40–0.62) *** | 2.40 (1.84–3.13) *** | 0.31 (0.25–0.40) *** | 0.25 (0.19–0.32) *** | 0.49 (0.40–0.59) *** | 1.37 (1.01–1.85) * |
| *Sex* | | | | | | | |
| Male (ref) | 1.00 | 1.00 | 1.00 | 1.00 | 1.00 | 1.00 | 1.00 |
| Female | 1.03 (0.93–1.14) | 1.26 (1.05–1.52) * | 1.31 (1.04–1.63) * | 1.18 (0.97–1.43) | 1.13 (0.93–1.38) | 1.56 (1.34–1.83) *** | 0.53 (0.41–0.68) *** |
| *Age group* | | | | | | | |
| 18–24 | 0.90 (0.76–1.06) | 1.06 (0.79–1.42) | 1.11 (0.78–1.56) | 0.95 (0.69–1.31) | 0.90 (0.63–1.28) | 0.63 (0.48–0.83) *** | 1.76 (1.24–2.51) ** |
| 25–64 (ref) | 1.00 | 1.00 | 1.00 | 1.00 | 1.00 | 1.00 | 1.00 |
| 65+ | 0.82 (0.73–0.93) ** | 0.94 (0.76–1.18) | 0.77 (0.58–1.02) | 1.48 (1.20–1.84) *** | 2.25 (1.83–2.78) *** | 2.70 (2.27–3.22) *** | 1.12 (0.83–1.51) |
| *Social grade* | | | | | | | |
| Higher social grade (ref) | 1.00 | 1.00 | 1.00 | 1.00 | 1.00 | 1.00 | 1.00 |
| Lower social grade | 0.57 (0.52–0.63) *** | 0.72 (0.60–0.87) *** | 0.91 (0.73–1.14) | 0.92 (0.76–1.12) | 1.14 (0.94–1.38) | 0.83 (0.71–0.97) * | 1.22 (0.95–1.56) |

*** = $p < 0.001$, ** = $p < 0.01$, * $p < 0.05$.

In general, by waves 2 and 3, respondents were less likely than in wave 1 to report worrying about social distancing in green spaces, fear for health when outdoors, a member of their household (or themselves) being at risk of severe consequences of COVID-19, or using an outdoor space at home as barriers to visiting green spaces (Table 3). These were quite substantially reduced odds, typically 0.40 or 0.50. In contrast, by wave 3 respondents were more likely than in wave 1 to report busy green spaces and/or a lack of interest in visiting green spaces as barriers to visits (Table 3). The odds of reporting busy green spaces as a barrier were substantially increased (OR: 2.40 (95% CI: 1.84–3.13)).

### 3.3. Variation in Green Space Visits and Barriers to Visits by Sex

There were no significant differences between male and female respondents in the likelihood of reporting visits to green spaces. However, female respondents were more likely than males to report three barriers to green space visits: being worried about social distancing in green space, green spaces being too busy, and using an outdoor space at home (Table 3). Odds for females reporting these, relative to males, were typically around 1.30. Female respondents were less likely than males to report a lack of interest as a barrier to visiting (OR: 0.53 (95% CI: 0.41–0.68)).

### 3.4. Variation in Green Space Visitation and Barriers by Age

Respondents aged 65+ were somewhat less likely to have visited green spaces in the last 4 weeks than those aged 25–64 (Table 3). This older group was also more likely to report fear for their health when outdoors, that they or a member of their household were at risk of severe consequences of COVID-19, and that they were using an outdoor space at home instead as barriers to visiting. These were the strongest associations seen in the models—the OR for using green spaces at home relative to the mid-age group was 2.70 (95% CI: 2.27–3.22), for example. In contrast, younger respondents were less likely to report using an outdoor space at home as a reason for not visiting green spaces. This age

group was also more likely to report not being interested in visiting green spaces, with a relatively large odds ratio of 1.76 (95% CI: 1.24–2.51).

### 3.5. Variation in Green Space Visitation and Barriers by Social Grade

Respondents in the lower social grade group were less likely than those in the higher grade group to have visited green spaces in the last 4 weeks (Table 3). The lower grade group were also less likely to report being worried about social distancing in green spaces and/or using an outdoor space at home instead as a barrier to visiting green spaces. These associations were relatively modest.

### 3.6. Change over Time in Associations

The addition of interaction terms to the models suggested several significant shifts over time (i.e., between waves) in the association between the socio-demographic variables and both visits and reporting barriers. Details of models with significant interactions are provided in Tables S2 and S3.

The association between visiting green spaces and social grade differed significantly between waves. This was the only socio-demographic variable to show a significant between-wave change in association with visits. The predicted probability plot (Figure 3—panel A) shows that, whilst the likelihood of visiting increased over time for both social grades, the increase was much sharper between waves 1 (April 2020) and 2 (November 2020) for higher social grades, followed by a more modest increase between waves 2 and 3 (April 2021). In contrast, the increase was relatively constant, wave to wave, for those in lower social grades. The result of these differences was an increased socio-economic inequality in visits in wave 3 compared to wave 1. The association between sex and reporting green spaces as being too busy to visit also differed significantly between waves. Figure 3—panel B suggests a reduction in the difference between men and women such that, by wave 3, the difference is lost.

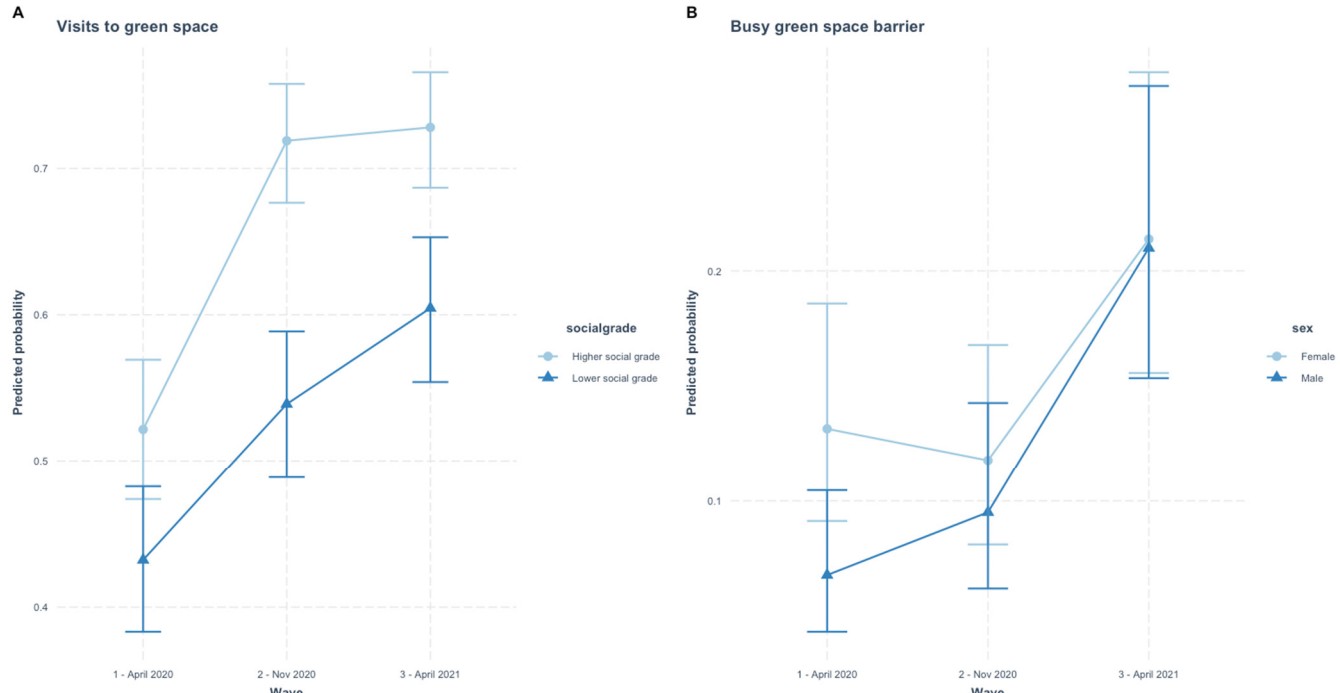

**Figure 3.** Predicted probability from logistic regression models with significant interaction for (**A**) visits to green space by wave and social grade and (**B**) the barrier 'green spaces are too busy' by wave and sex.

Whilst those were the only significant interactions involving social grade or sex, there were five significant interactions with age group (Figure 4). Figure 4—panel A suggests that whilst both 18–24 and 25–64 years old became less likely to report difficulty social distancing in green spaces as a barrier to use, respondents aged 65+ reporting that barrier remained relatively constant. A dip at wave 2 in reporting busy green spaces as a barrier to use by the youngest age group, and an overall steeper rise through time by the oldest age group probably explains the significant interaction (Figure 4—panel B). There were substantial falls for all age groups in reporting 'fear for my health' as a barrier to green space visits, but the fall was furthest and sharpest for the youngest age group, whilst it reduced between waves 2 and 3 for the older age groups (Figure 4—panel C).

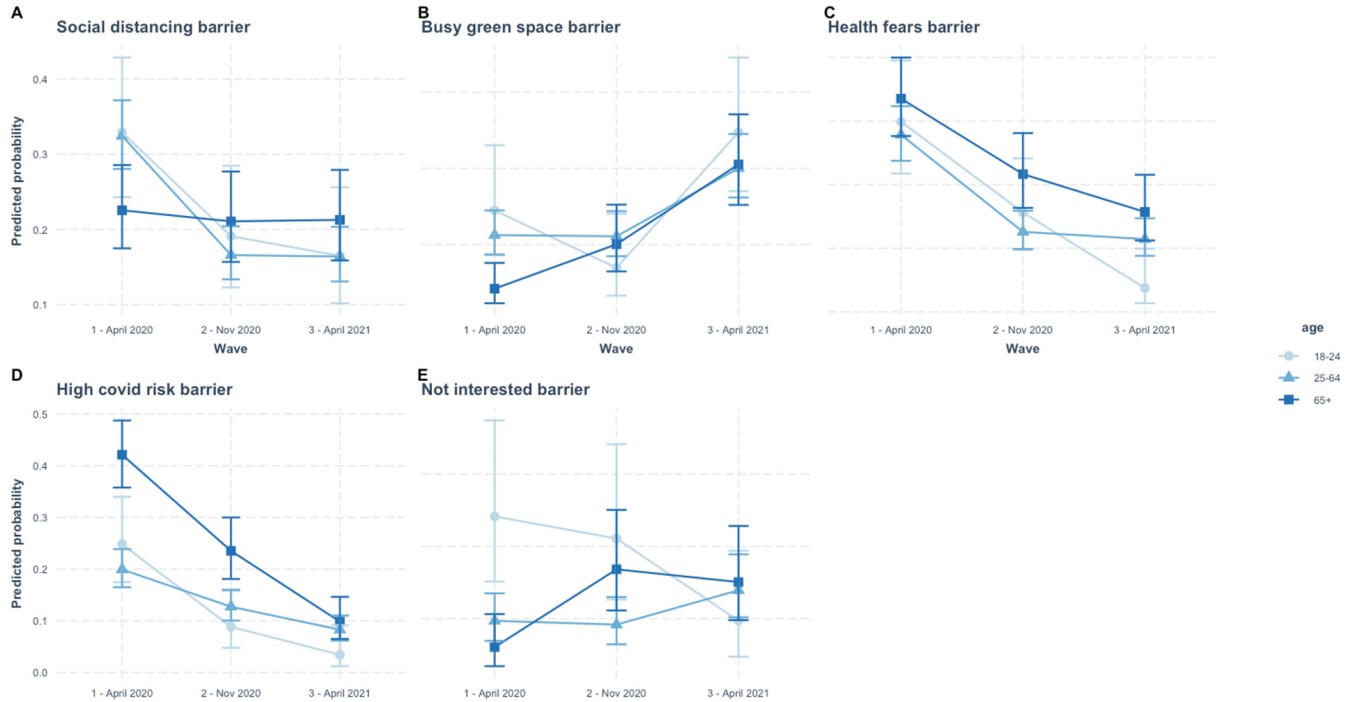

**Figure 4.** Predicted probabilities from logistic regression models with significant interactions between wave and age-group for: (**A**) the barrier 'I am worried I will not be able to socially distance', (**B**) the barrier 'green spaces are too busy', (**C**) the barrier 'I fear for my health when I go outdoors', (**D**) the barrier 'I/a member of my household is at higher risk of being severely affected by coronavirus', and (**E**) the barrier 'I am not interested in visiting green spaces'.

Reporting high risk of severe effects from COVID-19 as a barrier to the use of green spaces fell over time among all age groups, but a substantially higher starting level among those age 65+ and a relatively steeper decline produced a significant interaction (Figure 4—panel D). Finally, there were quite complex changes in the relationship between age groups and reporting a lack of interest in visiting green spaces (Figure 4—panel E). Perhaps the strongest signal from these was that the lack of interest fell sharply among the youngest age group, in contrast to rises among the older age groups.

## 4. Discussion

This study aims to describe the variation and change over time in green space visits and reported barriers to visiting during the COVID-19 pandemic in the UK. We drew on three waves of repeat cross-sectional data from April 2020, November 2020, and April 2021.

Visits to green spaces during the 4 weeks prior to the survey increased across the three waves—from 49% of respondents in April 2020 to 68% a year later. This increase was echoed in the regression results, confirming a significant association between survey wave and visiting. There were no other directly comparable surveys (in terms of timing) of green

space use in the UK, but several other cohort studies and surveys did take place. They largely echo our findings, suggesting substantial increases in visits to natural environments well beyond what had been achieved in the years prior to the pandemic [29,41,42]. If we assume that contact with nature is generally good for health and wellbeing, this marks a positive impact of COVID-19. However, the extent of the increase in visits shown by our data is partly driven by the timing of the surveys. We were fortunate to conduct wave 1 during the strictest lockdown the UK experienced. Outdoor exercise was strongly limited during this time. The data we present here capture visits to green spaces, but measures of time spent there point to an inevitable reduction as a consequence of the restrictions. By waves 2 and 3, restrictions on outdoor exercise were reduced and this probably partly explains the sharp rise [37]. It will be important to continue monitoring visits to green spaces as the pandemic wanes to see if a permanent shift in behaviour has taken place.

Although the proportion of people visiting green spaces increased, socio-economic inequality in visits also rose and this raises concerns about environmental justice. The environment itself did not alter, but behavioural response to it did. Throughout the study period, higher social grade respondents were more likely to have visited green spaces than the lower social grade respondents. The gap between these groups in terms of the predicted probability of visiting rose by 33% over time. Several other studies have noted socio-economic inequalities in the use of green spaces both pre- and inter-pandemic [19,41–44]. One likely contributor is inequality in access—a classic issue of environmental injustice. For example, a study found that British people with an annual household income lower than £15,000 were less likely to live within a 5-min walk of green spaces, to live somewhere where the streets are green, and to report good walking routes where they live, compared to households with an annual income of £35,000 or £70,000 [41]. However, another contributor could be the risk of infection when using green spaces. A recent study found that the boroughs in London with a higher risk of infection shared common characteristics. These included the low accessibility of green spaces, high covid case concentration, and high vulnerability to virus transmission (calculated using the Indices of Multiple Deprivation) [45]. Lower social grade respondents may have felt at more risk of contracting coronavirus when visiting neighbourhood green spaces compared to higher social grade respondents.

It is difficult to directly compare countries' changes in green space use and barriers to using green spaces during the COVID-19 pandemic. This is primarily due to international differences in the nature and timing of mobility restrictions at different stages of the pandemic and also to the dates of survey data collection in relation to these. Despite limitations to direct comparison, the evidence does suggest that there were substantial before and after changes in green space use in many countries following COVID-19, but the nature of the changes differed. Decreases in use were described in Saudi Arabia [46], Italy, and Spain [47], for example, whilst increases were described in Belgium [48] and Norway [49]. In New York, equal numbers of respondents reported that they increased (15%) and decreased (14%) their green space visits during the pandemic, which was influenced by Covid-related barriers (N = 1145). Individuals with greater concerns about crowded green spaces and lack of social distancing visited green spaces less often during the pandemic compared to before, while those who considered green spaces to be more important for their health visited more frequently [28]. Our study may be unique in having assessed changes in green space use at three different time points through the pandemic and in finding that increases in visits were sustained but COVID-19 specific barriers were not important in later waves.

Our study focused largely on barriers to visiting green spaces that could plausibly be created and mitigated by the progress of the pandemic. Results suggested a shift in the population's perceptions of risk. Being worried about social distancing, fearing for health, and respondents perceiving a household member or themselves as at risk of being severely affected by COVID-19 were all less likely to be reported by respondents in waves 2

and 3 compared to wave 1. This echoes studies from other countries. Research undertaken in Canada in 2020, for example, reported that some participants noted that seeing more people outside coincided with the easing of restrictions. They also stated that people viewed outdoor activities as permissible because it was occurring outdoors instead of indoors, and there was less risk of spreading coronavirus [50]. The relationships between reported barriers also shed some light on these changes. For example, by wave 3, respondents were more likely to report green spaces being too busy as a barrier than they had been in wave 1. This corroborates our finding that visits to green spaces increased over the year; the spaces really were busier. Furthermore, by wave 3, respondents were less likely to report being worried about social distancing as a barrier to using green spaces than in wave 1. This suggests concerns over crowding may be more connected to accessing the green space and having an enjoyable experience than worries about getting too close to others outdoors. This finding links with several other studies from around the world [32,51,52]. Research undertaken in Palestine, for example, found that respondents were more likely to visit green spaces alone, but less likely to 'relax' or 'socialise' in green spaces after the pandemic occurred compared to before [51]. It is possible that encouraging more people into green spaces had the unintended consequence of putting off others. Since these are repeat cross-sectional rather than panel data, we were unable to assess whether/who stopped visiting as a result.

Overall, the reduction in reporting of barriers relating to health and contracting COVID-19 may be due to both increased knowledge/understanding of the risk of contracting COVID-19 outdoors and progress with vaccination from December 2020 in the UK [52–54]. In late 2020 and early 2021, for example, studies identified a very low likelihood of COVID-19 transmission outdoors and these were widely reported in the UK [54–56]. However, both this new knowledge and the beginnings of the UK's vaccination programme would have likely only impacted the barriers reported in wave 3 and these were similar to those reported in wave 2. Further evidence that the public understood COVID-19 risks quite well comes from between-group differences in reporting barriers. The oldest group (65+) were most likely to report fearing for their health when outdoors and avoiding green spaces because they or a member of their household were at high risk of being severely affected by COVID-19. Given that those aged 80 years or older were seventy times more likely to die following a positive COVID-19 test compared with those under 40 [57], it seems reasonable older respondents reported these barriers to a greater extent.

Although we focused on barriers plausibly related to COVID-19, our inclusion of 'lack of interest' as a barrier revealed important trends. Across the whole year of study, respondents aged 18–24 years old were more likely than other age groups to report a lack of interest as a barrier to visiting green spaces. This echoes other research, including studies in England before and during the pandemic, which found that younger age groups (16–34 and 16–24) were more likely to report a lack of interest than older age groups [19,58]. However, in our study, there was a sharp reduction during the year in this lack of interest among young people, and one somewhat in contrast to the older age groups. An apparent increase in interest among young adults could be connected to the relaxing of restrictions from the end of March 2021. The change in restrictions meant outdoor spaces provided the only location for socialising with non-household members [37]. Perhaps younger groups became oriented to green spaces as a place to meet and socialise [59].

*Strengths and Limitations*

The study had several strengths including a nationally representative sample of the UK population when weightings were applied. To the best of our knowledge, our study provided the only data covering the UK population's change in green space use and barriers over one year of the COVID-19 pandemic starting in the period of maximum restrictions. The breadth of data enabled us to explore both changes in visits over time and reasons for not visiting. The latter is particularly poorly covered in the wider literature.

However, the survey data were self-reported and therefore liable to the usual biases, including recall and confirmation bias. The study was also repeated cross-sectionally rather than in a panel. The changes we observed and explored were therefore at a population or group level, not within individuals. There were some small numbers when exploring sub-groups who exhibited particular behaviours or attitudes. Data on the youngest group were particularly prone to this problem and it seems likely that bias will have been introduced. These problems highlight the need for further research into barriers to green space use for the wider population, as well as more specifically for young adults across the UK. The information on barriers was somewhat crude and we would encourage further research to investigate barriers to green space use in more detail, perhaps through qualitative methods or open-ended survey questions. In particular, the survey data on barriers analysed for this paper focused specifically on the influence of the COVID-19 pandemic on green space use. There may have been other barriers associated with the pandemic but not captured here, such as changes in behaviours imposed by home schooling and home working. These are also likely to have been unequal.

## 5. Conclusions

At a time when the COVID-19 pandemic is ongoing and restrictions are continuing to change, our study provides novel findings on how the UK population's visits to green spaces, and barriers to visiting, changed over the first year of the pandemic. Inequalities in the use of green spaces by social grade were sustained and widened over the year, emphasising that environmental injustices were exacerbated. There is a need for further investigation and action to ensure equal access to good quality green spaces for all demographic groups and communities. The reporting of Covid-related barriers to green space use, such as fear for their health and worrying about social distancing, fell over time, in parallel perhaps with greater knowledge of the virus, how it spreads, and perceptions of risk. Levels of disinterest in visiting green spaces also changed over the year, particularly for 18–24 years old; the pandemic seems to have made changes to their perceptions of green spaces. Our findings suggest that, despite the benefits to health and wellbeing that green spaces use can provide, there are still barriers in place that restrict some of the population from using them. These findings also indicate that the barriers in place are not felt equally across socio-demographic groups, particularly by age. The pandemic has affected green space use in both positive and negative ways; it will be important to determine if these are permanent changes. We encourage further research exploring young people's use and perceptions of green spaces, with a particular focus on interest in green spaces and how the patterns relating to a 'lack of interest' may change as all lockdown restrictions are removed. Our study has highlighted that inequalities in the use of green spaces remain and future research should be focused on reasons for a lack of green space use that is not related to COVID-19, which remains a key gap in the existing literature.

**Supplementary Materials:** The following are available online at https://www.mdpi.com/article/10.3390/land11040503/s1, Table S1: Weighted counts and percentages of demographic variables, Table S2: Significant interaction results by wave and socio-demographic variables; Predicted Probabilities (95% Confidence Intervals>), Table S3: Significant interaction results by wave and socio-demographic variables, each interaction model was adjusted for wave, sex, age, and social grade; Interaction Odds Ratios (95% Confidence Intervals).

**Author Contributions:** Conceptualization, H.B., J.R.O. and R.M.; methodology, H.B., J.R.O. and R.M.; software, H.B.; validation, H.B., J.R.O. and R.M.; formal analysis, H.B.; data curation, H.B.; writing—original draft preparation, H.B.; writing—review and editing, H.B., J.R.O. and R.M.; visualization, H.B. All authors have read and agreed to the published version of the manuscript.

**Funding:** The authors (H.B., J.R.O. and R.M.) are part of the Places and Health Programme at the MRC/CSO Social and Public Health Sciences Unit (SPHSU), University of Glasgow, supported by the Medical Research Council (MC_UU_00022/4) and the Chief Scientist Office (SPHSU19). H.B. is also

funded by a Medical Research Council and University of Glasgow College of Medical, Veterinary and Life Sciences PhD studentship (MC_ST_U18004).

**Institutional Review Board Statement:** Ethical review and approval were waived for this study due to the data collection being conducted by YouGov, who use a double opt-in procedure across all Panels. All data were anonymized.

**Informed Consent Statement:** Informed consent was obtained by YouGov from all subjects involved in the Panel.

**Data Availability Statement:** We have made our research dataset publicly available (Datacite DOI: 10.5525/gla.researchdata.1038 and 10.5525/gla.researchdata.1091—embargoed until 31 October 2023).

**Conflicts of Interest:** The authors declare no conflict of interest.

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
