# Peer review of "Green Space Visits and Barriers to Visiting during the COVID-19 Pandemic: A Three-Wave Nationally Representative Cross-Sectional Study of UK Adults"

_land, doi:10.3390/land11040503_

Round 1

Reviewer 1 Report

The manuscript presents results of a representative three-wave survey of the UK population regarding green space visits and barriers to visits with the emphasis on the relationship between visits and socio-demographic variables in three points of time. The research method is well implemented and data properly analysed. 

However, some points need to be addressed before the manuscript would be ready for publication. First of all the manuscript does not provide any theoretical background or analytical framework (in my opinion the topic and results fit well environmental justice), as well as sufficient literature coverage against which the results could be compared. This is especially visible in the discussion part. In the discussion, results should be compared to similar literature addressing green space visiting behaviours during the COVID-19 pandemic. There have been many publications in the past year or so dealing with this topic. 

The conclusions part seems more like repeating the results and should be improved.  

Another point is about the barriers or reasons why people visited green space less or not at all. It cannot be changed now but strikes me that the reasons were only related to fear of contracting the disease while visiting green space. Also, all the answers were predefined and there was no possibility to provide other answers. In my opinion, there should have been left that possibility to be able to learn from respondents' experiences. What comes to my mind is very much a sex-related obstacle. Studies reported that women had less time for visiting green space, especially during lockdown, due to combination of home office and homeschooling which is a significant obstacle that was not taken into consideration in this survey. However, like I said it is not possible to change it now. 

Author Response

Thank you for your comments.

Response to Reviewer 1 Comments

The manuscript presents results of a representative three-wave survey of the UK population regarding green space visits and barriers to visits with the emphasis on the relationship between visits and socio-demographic variables in three points of time. The research method is well implemented and data properly analysed. However, some points need to be addressed before the manuscript would be ready for publication.

Response:

We thank the reviewer for their positive comments on the suitability of the implementation of the methods and analysis.

First of all the manuscript does not provide any theoretical background or analytical framework (in my opinion the topic and results fit well environmental justice).

Response:

The theoretical background and analytical framework of this study was concerned with exploring differences in greenspace access and use according to socio-economic characteristics from an environmental justice and inequalities perspective. We have provided the following additional text within the introduction:

“This paper is concerned with  measuring green space access from an environmental justice and inequalities perspective by exploring differences according to socio-economic characteristics at a neighbourhood and individual level [22,23]. This paper aimed to provide insights that may add to the existing environmental justice framework as applied to green space, with a focus on use and barriers rather than access and quality [24]. Whereas much environmental justice research explores inequalities in exposure to pathogenic environments, our work focuses on justice in both access to, and use of, a salutogenic environment. We explore how the Covid-19 pandemic may have exacerbated environmental injustice, particularly for those of differing social grades/occupations. It is important to explore environmental injustice alongside the “equigenic hypothesis”[25], which suggests that some environments (such as natural environments) can support the health of the less advantaged as much as, or even more than, the more advantaged [26,27]. By exploring green space visits and barriers during the pandemic, we can strengthen understanding of who is not using green spaces, why, and how the pandemic impacted upon existing inequalities.” (Page 2, lines 78-92)

We have also provided further text within the discussion:

“Although the proportion of people visiting green space increased, socio-economic inequality in visits also rose and this raises concerns about environmental justice. It’s not that the environment itself altered, but that behavioural response to it did.” (Page 12/13, lines 392-398)

“One likely contributor is inequality in access – a classic issue of environmental injustice.” (Page 13, lines 403-404)

“Inequalities in use of green space by social grade were sustained and widened over the year, emphasising that environmental injustices were exacerbated.” (Page 15, lines 586-588)

… as well as sufficient literature coverage against which the results could be compared. This is especially visible in the discussion part. In the discussion, results should be compared to similar literature addressing green space visiting behaviours during the COVID-19 pandemic. There have been many publications in the past year or so dealing with this topic. 

Response:

We have expanded our discussion to provide further context of our results in the perspective of other global literature, particularly literature that focuses on green space visiting behaviour during the Covid-19 pandemic:

“It is difficult to directly compare between countries the changes in green space use and barriers to using green space during the Covid-19 pandemic. This is primarily due to international differences in the nature and timing of mobility restrictions at different stages of the pandemic, and also to the dates of survey data collection in relation to these. Despite limitations to direct comparison, evidence does suggest that there were substantial before and after changes in park use in many countries following Covid-19, but the nature of the changes differed. Decreases in use were described in Saudi Arabia [49], Italy and Spain [42], for example, whilst increases were described in Belgium [50] and Norway [41]. In New York, equal numbers of respondents reported that they increased (15%) and decreased (14%) their green space visits during the pandemic, which was influenced by Covid-related barriers (N=1,145). Individuals with greater concerns about crowded green space and lack of social distancing visited green space less often during the pandemic compared to before, while those who considered green space to be more important for their health visited more frequently [28]. Our study may be unique in having assessed changes in green space use at three different time points through the pandemic and in finding that increases in visits were sustained but Covid-19 specific barriers were not important in later waves.” (Page 13, lines 416-432)

“This echoes studies from other countries. Research undertaken in Canada in 2020, for example, reported that some participants noted that seeing more people outside coincided with the easing of restrictions. They also stated that people viewed outdoor activities as permissible because it was occurring outdoors instead of indoors, and there was less risk of spreading coronavirus [51].” (Page 13, lines 439-443)

This findings links with several other studies from around the world [32,51,52]. Research undertaken in Palestine, for example, found that respondents were more likely to visit green spaces alone, but less likely to ‘relax’ or ‘socialise’ in green space, after the pandemic occurred compared to before [52]. It is possible that encouraging more people into green spaces had the unintended consequence of putting off others. Since these are repeat cross-sectional rather than panel data, we were unable to assess whether / who stopped visiting as a result.” (Page 13/14, lines 451-518)

The conclusions part seems more like repeating the results and should be improved.  

Response:

We thank the reviewer for this comment and have added to the conclusion section, ensuring the messaging is improved and it is distinct from the results section:

“These findings also indicate that the barriers in place are not felt equally across socio-demographic groups, particularly by age. The pandemic has affected green space use in both positive and negative ways; it will be important to determine if these are permanent changes. We encourage further research exploring young people’s use and perceptions of green space, with a particular focus on interest in green space and how the patterns relating to a ‘lack of interest’ may change as all lockdown restrictions are removed. Our study has highlighted that inequalities in use of green space remain, and future research should be focused on reasons for a lack of green space use that are not related to Covid-19, which remains a key gap in the existing literature.” (Page 15, lines 597-605)

Another point is about the barriers or reasons why people visited green space less or not at all. It cannot be changed now but strikes me that the reasons were only related to fear of contracting the disease while visiting green space. Also, all the answers were predefined and there was no possibility to provide other answers. In my opinion, there should have been left that possibility to be able to learn from respondents' experiences. What comes to my mind is very much a sex-related obstacle. Studies reported that women had less time for visiting green space, especially during lockdown, due to combination of home office and homeschooling which is a significant obstacle that was not taken into consideration in this survey. However, like I said it is not possible to change it now. 

Response:

The reviewer has highlighted a key limitation in our study design that barriers to using greenspace across the three waves were pre-defined and open-ended questions were not included. Our survey did collect a number of other Covid-related barriers to using greenspace but were out of scope of the key aims and objectives of this study, we made a decision to focus on the Covid-related barriers in this paper to correspond with the nature of this Special Issue, focusing on changes in use and perceptions of green space during the Covid-19 pandemic. The aim of the three waves of survey data being collected was also to specifically explore how the pandemic, and lockdowns, were influencing the population’s use, experiences and barriers to green space.

We have added this limitation to the manuscript and propose other key barriers that we did not measure here: 

“The information on barriers was somewhat crude and we would encourage further research to investigate barriers to green space use in more detail, perhaps through qualitative methods or open-ended survey questions. In particular, the survey data on barriers analysed for this paper focused specifically on the influence of the Covid-19 pandemic on green space use. There may have been other barriers associated with the pandemic but not captured here, such as changes in behaviours imposed by home schooling and home working. These are also likely to have been unequal.” (Page 14/15, lines 565-581)

We now also recommend studies explore other barriers to green space use that are not related to Covid-19:

“We encourage further research exploring young people’s use and perceptions of green space, with a particular focus on interest in green space and how the patterns relating to a ‘lack of interest’ may change as all lockdown restrictions are removed. Our study has highlighted that inequalities in use of green space remain, and future research should be focused on reasons for a lack of green space use that are not related to Covid-19, which remains a key gap in the existing literature.” (Page 15, lines 600-605)

Reviewer 2 Report

Although well structured and with the adequately applied methodology, the article would benefit from introducing/comparing the results of similar studies conducted abroad. The main aim of the analysis could be more specified, as well as the discussion which might include some recommendations for the further research, comparisons and improvements regarding the detected barriers. Furthermore, the results of the survey could be displayed in a graphically appropriate way (e.g. charts, pictograms etc).

Author Response

Thank you for your comments.

Response to Reviewer 2 Comments

Comments:

Although well structured and with the adequately applied methodology, the article would benefit from introducing/comparing the results of similar studies conducted abroad.

Response:

We thank the reviewer for this comment, which was similar to Reviewer 1. In response to both comments we have provided further international context in the discussion of our results:

“Studies in the UK, for example, found that the mental health of adults followed an ‘up and down’ cycle coinciding with periods of national lockdown and high Covid-19 case rates [5,6]. This negative impact on mental health has been felt internationally [7–10]. For example, in a study undertaken in Germany, perceived stress increased across all age groups over the six weeks following the mid-March 2020 Covid-19 outbreak [7]. The exact mental health burden of the pandemic is still too early to measure, with many countries experiencing new waves of virus cases [8].” (Page 2, lines 52-59)

“A recent study in Korea found that individuals with decreased visits to green space between September-December 2020 had 2.06 higher odds of probable major depression at the time of the survey compared to those whose visits had increased or stayed the same [31]. This emphasises the importance of researching the barriers to green space use during the pandemic by sociodemographic characteristics, and exploring how barriers changed over its first year.” (Page 3, lines 107-112)

“It is difficult to directly compare between countries the changes in green space use and barriers to using green space during the Covid-19 pandemic. This is primarily due to international differences in the nature and timing of mobility restrictions at different stages of the pandemic, and also to the dates of survey data collection in relation to these. Despite limitations to direct comparison, evidence does suggest that there were substantial before and after changes in park use in many countries following Covid-19, but the nature of the changes differed. Decreases in use were described in Saudi Arabia [49], Italy and Spain [42], for example, whilst increases were described in Belgium [50] and Norway [41]. In New York, equal numbers of respondents reported that they increased (15%) and decreased (14%) their green space visits during the pandemic, which was influenced by Covid-related barriers (N=1,145). Individuals with greater concerns about crowded green space and lack of social distancing visited green space less often during the pandemic compared to before, while those who considered green space to be more important for their health visited more frequently [28]. Our study may be unique in having assessed changes in green space use at three different time points through the pandemic and in finding that increases in visits were sustained but Covid-19 specific barriers were not important in later waves.” (Page 13, lines 416-432)

“This echoes studies from other countries. Research undertaken in Canada in 2020, for example, reported that some participants noted that seeing more people outside coincided with the easing of restrictions. They also stated that people viewed outdoor activities as permissible because it was occurring outdoors instead of indoors, and there was less risk of spreading coronavirus [51].” (Page 13, lines 439-443)

This findings links with several other studies from around the world [32,51,52]. Research undertaken in Palestine, for example, found that respondents were more likely to visit green spaces alone, but less likely to ‘relax’ or ‘socialise’ in green space, after the pandemic occurred compared to before [52]. It is possible that encouraging more people into green spaces had the unintended consequence of putting off others. Since these are repeat cross-sectional rather than panel data, we were unable to assess whether / who stopped visiting as a result.” (Page 13/14, lines 451-518)

The main aim of the analysis could be more specified, as well as the discussion which might include some recommendations for the further research, comparisons and improvements regarding the detected barriers.

Response:

Thank you for this comment, we have now specified the main aim of the analysis and also restructured the statistical analyses section to ensure that it is clear and easy to follow (Page 4, lines 195-216).

The aims have been updated:

Further research is required to describe whether the increased visit numbers were sustained as the Covid-19 pandemic continued, how barriers to visiting changed following subsequent easing and reintroduction of Covid-19 measures, and inequalities in these. There is a lack of literature in this area, with no studies to date focusing on change in green space use and barriers in the UK over the first year of the pandemic. It is important to explore this gap, both in order to understand how the experience of UK relates to that in other countries, and also to promote resilient green spaces and wider societies in the face of future pandemics [32]. The research objectives of this paper were therefore:

  1. To describe variation in green space visiting during the Covid-19 pandemic.
  2. To describe reported barriers to green space visits during the Covid-19 pandemic.
  3. To explore variation in green space visits and barriers over time and by sex, age and socio-economic position.” (Page 3, lines 114-126)

The barriers included here with specific to Covid-19 and pre-specified. We agree with the reviewer that other barriers should be explored and now highlight a recommendation for further qualitative or open-ended surveys to collect this data, for both Covid-19 and inequalities research.:

“The information on barriers was somewhat crude and we would encourage further research to investigate barriers to green space use in more detail, perhaps through qualitative methods or open-ended survey questions. In particular, the survey data on barriers analysed for this paper focused specifically on the influence of the Covid-19 pandemic on green space use. There may have been other barriers associated with the pandemic but not captured here, such as changes in behaviours imposed by home schooling and home working. These are also likely to have been unequal.” (Page 14/15, lines 565-581)

We have also added recommendations for further research in the conclusions section:

“We encourage further research exploring young people’s use and perceptions of green space, with a particular focus on interest in green space and how the patterns relating to a ‘lack of interest’ may change as all lockdown restrictions are removed. Our study has highlighted that inequalities in use of green space remain, and future research should be focused on reasons for a lack of green space use that are not related to Covid-19, which remains a key gap in the existing literature.” (Page 15, lines 600-605)

Furthermore, the results of the survey could be displayed in a graphically appropriate way (e.g. charts, pictograms etc).

Response:

We have now included two additional figures (Figure 1, page 7 & Figure 2, page 8) to display the change in the percentage of respondents visiting green space, or not visiting green space, over the three survey waves and by socio-demographic characteristics.

Reviewer 3 Report

Over the course of a year in the UK during the pandemic, this study looked at green space visits, barriers to visiting, and inequalities in both. In April 2020, November 2020, and April 2021, YouGov conducted three waves of a nationally representative cross-sectional poll (N=6,713). Visits to green space were reported, as well as perceived impediments, such as those likely connected to Covid-19 risk, for individuals who did not visit or who visited infrequently. As lockdown limits were removed, green space visits grew over the year, with 68 percent of respondents reporting green space trips in April 2021, up from 49 percent in April 2020. The study contains some interesting data, but both the introduction and discussions must include recent publications. In particular, the introduction is too short, and the research gap needs to be highlighted. The findings of this study should be compared to those of previous research conducted in the United Kingdom and internationally (see articles published in Urban Forestry and Urban Greening).

Author Response

Thank you for your comments.

Response to Reviewer 3 Comments

Over the course of a year in the UK during the pandemic, this study looked at green space visits, barriers to visiting, and inequalities in both. In April 2020, November 2020, and April 2021, YouGov conducted three waves of a nationally representative cross-sectional poll (N=6,713). Visits to green space were reported, as well as perceived impediments, such as those likely connected to Covid-19 risk, for individuals who did not visit or who visited infrequently. As lockdown limits were removed, green space visits grew over the year, with 68 percent of respondents reporting green space trips in April 2021, up from 49 percent in April 2020.

The study contains some interesting data, but both the introduction and discussions must include recent publications. In particular, the introduction is too short, and the research gap needs to be highlighted. The findings of this study should be compared to those of previous research conducted in the United Kingdom and internationally (see articles published in Urban Forestry and Urban Greening).

Response:

We thank the reviewer for this comment, which was similar to Reviewer 1. In response to both comments we have provided further international context in the discussion of our results, including references published in key academic journals, such as Urban Forestry and Urban Greening:

“It is difficult to directly compare between countries the changes in green space use and barriers to using green space during the Covid-19 pandemic. This is primarily due to international differences in the nature and timing of mobility restrictions at different stages of the pandemic, and also to the dates of survey data collection in relation to these. Despite limitations to direct comparison, evidence does suggest that there were substantial before and after changes in park use in many countries following Covid-19, but the nature of the changes differed. Decreases in use were described in Saudi Arabia [49], Italy and Spain [42], for example, whilst increases were described in Belgium [50] and Norway [41]. In New York, equal numbers of respondents reported that they increased (15%) and decreased (14%) their green space visits during the pandemic, which was influenced by Covid-related barriers (N=1,145). Individuals with greater concerns about crowded green space and lack of social distancing visited green space less often during the pandemic compared to before, while those who considered green space to be more important for their health visited more frequently [28]. Our study may be unique in having assessed changes in green space use at three different time points through the pandemic and in finding that increases in visits were sustained but Covid-19 specific barriers were not important in later waves.” (Page 13, lines 416-432)

“This echoes studies from other countries. Research undertaken in Canada in 2020, for example, reported that some participants noted that seeing more people outside coincided with the easing of restrictions. They also stated that people viewed outdoor activities as permissible because it was occurring outdoors instead of indoors, and there was less risk of spreading coronavirus [51].” (Page 13, lines 439-443)

This findings links with several other studies from around the world [32,51,52]. Research undertaken in Palestine, for example, found that respondents were more likely to visit green spaces alone, but less likely to ‘relax’ or ‘socialise’ in green space, after the pandemic occurred compared to before [52]. It is possible that encouraging more people into green spaces had the unintended consequence of putting off others. Since these are repeat cross-sectional rather than panel data, we were unable to assess whether / who stopped visiting as a result.” (Page 13/14, lines 451-518)

We have also added more text and citations to the introduction and ensured that the research gap is highlighted by including the following text:

“Studies in the UK, for example, found that the mental health of adults followed an ‘up and down’ cycle coinciding with periods of national lockdown and high Covid-19 case rates [5,6]. This negative impact on mental health has been felt internationally [7–10]. For example, in a study undertaken in Germany, perceived stress increased across all age groups over the six weeks following the mid-March 2020 Covid-19 outbreak [7]. The exact mental health burden of the pandemic is still too early to measure, with many countries experiencing new waves of virus cases [8].” (Page 2, lines 52-59)

“There are considerable social, economic and gender inequalities in visits to green space; less advantaged populations are far less likely to do so [19–21]. Barriers to use of green spaces are socially and spatially unequal. Not everyone has good access to a suitable space, or the time and inclination to visit it. This paper is concerned with  measuring green space access from an environmental justice and inequalities perspective by exploring differences according to socio-economic characteristics at a neighbourhood and individual level [22,23]. This paper aimed to provide insights that may add to the existing environmental justice framework as applied to green space, with a focus on use and barriers rather than access and quality [24]. Whereas much environmental justice research explores inequalities in exposure to pathogenic environments, our work focuses on justice in both access to, and use of, a salutogenic environment. We explore how the Covid-19 pandemic may have exacerbated environmental injustice, particularly for those of differing social grades/occupations. It is important to explore environmental injustice alongside the “equigenic hypothesis”[25], which suggests that some environments (such as natural environments) can support the health of the less advantaged as much as, or even more than, the more advantaged [26,27]. By exploring green space visits and barriers during the pandemic, we can strengthen understanding of who is not using green spaces, why, and how the pandemic impacted upon existing inequalities.” (Page 2, lines 75-92)

“A recent study in Korea found that individuals with decreased visits to green space between September-December 2020 had 2.06 higher odds of probable major depression at the time of the survey compared to those whose visits had increased or stayed the same [31]. This emphasises the importance of researching the barriers to green space use during the pandemic by sociodemographic characteristics, and exploring how barriers changed over its first year.” (Page 3, lines 107-112)

“There is a lack of literature in this area, with no studies to date focusing on change in green space use and barriers in the UK over the first year of the pandemic. It is important to explore this gap, both in order to understand how the experience of UK relates to that in other countries, and also to promote resilient green spaces and wider societies in the face of future pandemics [32].” (Page 3, lines 117-121)

Round 2

Reviewer 3 Report

The paper has been improved.